# Three-year outcomes of the randomized phase III SEIPLUS trial of extensive intraoperative peritoneal lavage for locally advanced gastric cancer

Jing Guo[1,2,3,14], Aman Xu[4,14], Xiaowei Sun[3,14], Xuhui Zhao[5,14], Yabin Xia[6,14], Huamin Rao[7,14], Yaming Zhang[8], Rupeng Zhang[9], Li Chen[10], Tao Zhang[11], Gang Li[12], Hongtao Xu[13] & Dazhi Xu [1,2,3,14 ✉]

Whether extensive intraoperative peritoneal lavage (EIPL) after gastrectomy is beneficial to patients with locally advanced gastric cancer (AGC) is not clear. This phase 3, multicenter, parallel-group, prospective randomized study (NCT02745509) recruits patients between April 2016 and November 2017. Eligible patients who had been histologically proven AGC with T3/4NxM0 stage are randomly assigned (1:1) to either surgery alone or surgery plus EIPL. The results of the two groups are analyzed in the intent-to-treat population. A total of 662 patients with AGC (329 patients in the surgery alone group, and 333 in the surgery plus EIPL group) are included in the study. The primary endpoint is 3-year overall survival (OS). The secondary endpoints include 3-year disease free survival (DFS), 3-year peritoneal recurrence-free survival (reported in this manuscript) and 30-day postoperative complication and mortality (previously reported). The trial meets pre-specified endpoints. Estimated 3-year OS rates are 68.5% in the surgery alone group and 70.6% in the surgery plus EIPL group (log-rank p = 0.77). 3-year DFS rates are 61.2% in the surgery alone group and 66.0% in the surgery plus EIPL group (log-rank p = 0.24). The pattern of disease recurrence is similar in the two groups. In conclusion, EIPL does not improve the 3-year survival rate in AGC patients.

[1] Department of Gastric Surgery, Fudan University Shanghai Cancer Center, Shanghai, China. [2] Department of Oncology, Shanghai Medical College, Fudan University, Shanghai, China. [3] Department of Gastric Surgery, Sun Yat-sen University Cancer Center, Guangzhou, Guangdong, China. [4] Department of Gastrointestinal Surgery, The First Affiliated Hospital of Anhui Medical University, HeFei, Anhui, China. [5] Department of General Surgery, The First Affiliated Hospital of University of Science and Technology of China (Anhui Provincial Cancer Hospital), Hefei, Anhui, China. [6] Department of General Surgery, The First Affiliated Hospital of Wannan Medical College, Wuhu, Anhui, China. [7] Department of Abdominal Surgery, Jiangxi Provincial Cancer Hospital, Nanchang, Jiangxi, China. [8] Department of Surgical Oncology, Anqing Municipal Hospital, Anqing, Anhui, China. [9] Department of Gastric Surgery, Tianjin Medical University Cancer Institute and Hospital, Tianjin, Tianjin, China. [10] Department of General Surgery, The Second Affiliated Hospital of Zhejiang University School of Medicine, Hangzhou, Zhejiang, China. [11] Department of Gastrointestinal Surgery, Yuebei People's Hospital, Shaoguan, Guangdong, China. [12] Department of General Surgery, Jiangsu Cancer Hospital, Nanjing, Jiangsu, China. [13] Department of General Surgery, Lishui Municipal Central Hospital, Lishui, Zhejiang, China. [14]These authors contributed equally: Jing Guo, Aman Xu, Xiaowei Sun, Xuhui Zhao, Yabin Xia, Huamin Rao. ✉email: xudzh@shca.org.cn

For locally advanced gastric cancer (AGC), surgical resection is the cornerstone of treatment[1]. However, the 5-year overall-survival (OS) rates are still low, 56.9%, 37.4%, and 16.2% for AGC patients with stage pT3, pT4a, and pT4b after surgery, respectively[2]. Peritoneum is the most common site of recurrence after curative resection, especially for serosa-positive patients (pT3-4)[3–5]. The prognosis of patients with peritoneal metastasis (PM) is extremely dismal with median survival time less than 1 year[6,7].

Generally, PM is caused by free cancer cells detached from the serosal surface of the stomach and implanted on the intraperitoneal wall[8]. For example, intraoperatively primary tumor manipulation and lymphadenectomy often lead to cancer cells exfoliation[9,10]. Therefore, elimination of free cancer cells during surgery is a promising strategy to prevent PM.

As a practical procedure for reducing the risk of PM, extensive intraoperative peritoneal lavage (EIPL) is an approach in which the peritoneal cavity is repeatedly (10 times) washed with 1 L of warm normal saline after potentially curative gastrectomy[11]. In 2009, Kuramoto at el. first reported that EIPL plus intraperitoneal chemotherapy could reduce PM significantly in AGC patients[12]. Subsequently, four multicenter randomized controlled trials (RCT) from Japan (CCOG1102 RCT), Singapore (EXPEL RCT), and Brazil, and the current SEIPLUS RCT were conducted to assess the value of the EIPL as a prophylactic strategy for PM of locally AGC[13–16]. In addition to our study, CCOG1102 study has also been completed. It showed that EIPL did not improve survival or cancer recurrence[14]. However, the sample size of CCOG1102 study was only 314 cases, which was not enough for evaluating the OS of patients. Moreover, the stage of patients was relatively early, and the probability of peritoneal metastasis is small, which weakened the significance of CCOG1102 study[17–19].

Previously, we have published the early results of the SEIPLUS trial[16]. It showed that EIPL was a safe and simple procedure. Furthermore, we found that EIPL could decrease postoperative short-term complications[16]. In this work, we report the 3-year survival and disease recurrence outcomes of SEIPLUS trial. We show treatment with surgery plus EIPL does not improve the 3-year survival rate in AGC patients compared with surgery alone.

## Results

**Patient characteristics.** Between April 2016 and November 2017, 662 patients from 11 centers in China were preoperatively enrolled and randomly assigned (329 patients assigned to the surgery alone group and 333 assigned to the surgery plus EIPL group). According to histopathologic results, 112 patients (58 in the surgery alone group and 54 in the surgery plus EIPL group) were ineligible owing to T1, T2, or M1 disease. The CONSORT diagram was presented as Fig. 1. Finally, 550 patients were eligible to the inclusion criteria. All 662 patients randomized were included in our intention-to-treat (ITT) analysis. The baseline patient and tumor characteristics were well balanced between the two groups (Table 1). The multivariate Cox regression analyzed a nonsignificant HR of 0.95 (95% CI, 0.74–1.22, $p = 0.69$) comparing surgery alone with surgery plus EIPL (Supplementary Table S1). Blinded outcome assessment was performed throughout the study.

**Primary and secondary outcomes: overall survival, disease-free survival, and peritoneal recurrence-free survival.** After a median follow-up of 47.8 (95% CI 46.9–48.7) months, 122 (37.1%) patients in the EIPL group died compared with 123 (36.9%) patients in the surgery alone group. Ninety one (27.7%) patients in surgery alone group and 79 (23.7%) patients in the surgery plus EIPL group had disease relapse. The pattern of disease recurrence was similar in the two groups. The most common site of first relapse was the peritoneum (Table 2). Only five (0.8%) patients were lost to follow-up at the time of the final analysis (three in the surgery alone group and two in the EIPL group). Eleven (3.3%) patients in the surgery alone group and 22 (6.6%) in the surgery plus EIPL group died from causes other than disease recurrence and surgery complications. Five (1.5%) patients in the surgery alone group and no patient in the surgery plus EIPL group died from surgical complications.

Estimated 3-year OS rates were 68.5% (95% CI 63.5–73.5) in the surgery alone group and 70.6% (95% CI 65.7–75.5) in the surgery plus EIPL group. Estimated 3-year DFS rates were 61.2% (95% CI 55.9–66.5) in the surgery alone group and 66.0% (95% CI 60.9–71.1) in the surgery plus EIPL group. Estimated 3-year peritoneal recurrence-free survival rates were 66.6% (95% CI 61.5–71.7) in the surgery alone group and 70.0% (95% CI 65.1–74.9) in the surgery plus EIPL group. The OS (HR 0.97, 95% CI 0.75–1.24, log-rank $p = 0.77$), DFS (HR 0.87, 95% CI 0.69–1.10, log-rank $p = 0.24$) and peritoneal recurrence-free survival (HR 0.93, 95% CI 0.72–1.19, log-rank $p = 0.54$) in all 662 patients did not differ significantly between the surgery alone and surgery plus EIPL groups. Kaplan–Meier curves for OS, DFS,

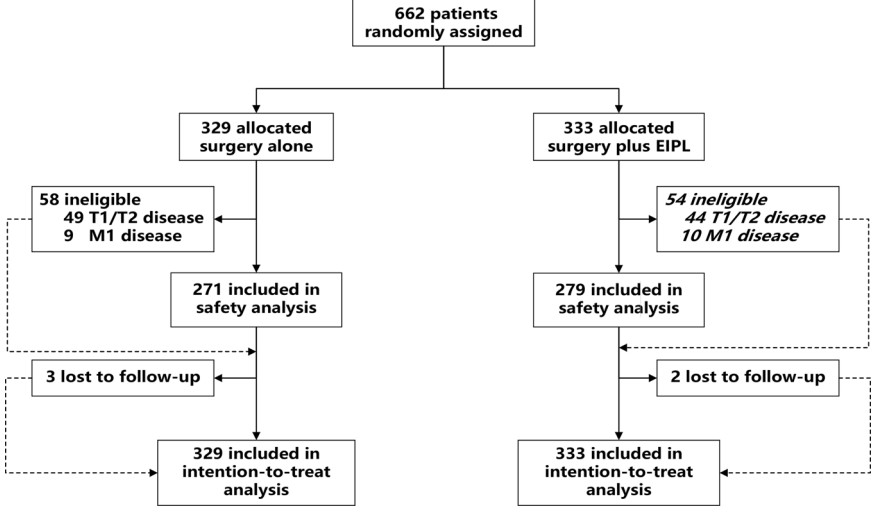

**Fig. 1 Trial CONSORT flow diagram.** EIPL extensive intraoperative peritoneal lavage.

**Table 1 Demographics and baseline characteristics.**

|  | Surgery alone (n = 329) | Surgery + EIPL (n = 333) |
|---|---|---|
| Age, mean (SD), y | 60.7 (10.7) | 60.7 (10.6) |
| Sex (n) |  |  |
| Male | 245 (74.5%) | 233 (70.0%) |
| Female | 84 (25.5%) | 100 (30.0%) |
| Smoking status (n) |  |  |
| Yes | 104 (31.6%) | 109 (32.7%) |
| No | 225 (68.4%) | 224 (67.3%) |
| BMI, mean (SD) | 22.0 (2.9) | 22.2 (3.1) |
| Tumor location (n) |  |  |
| Upper 1/3 | 100 (30.4%) | 97 (29.1%) |
| Middle 1/3 | 88 (26.7%) | 86 (25.8%) |
| Lower 1/3 | 130 (39.5%) | 140 (42.1%) |
| Total | 11 (3.4%) | 10 (3.0 %) |
| Tumor size, mean (SD), cm | 5.0 (3.1) | 5.0 (2.5) |
| Pathologic T stage (n) |  |  |
| T1/2 | 49 (14.9%) | 44 (13.2%) |
| T3 | 74 (22.5%) | 76 (22.8%) |
| T4 | 206 (62.6%) | 213 (64.0%) |
| Pathologic N stage (n) |  |  |
| N0 | 73 (22.2%) | 75 (22.5%) |
| N1 | 62 (18.8%) | 70 (21.0%) |
| N2 | 82 (24.9%) | 76 (22.8%) |
| N3 | 112 (34.1%) | 112 (33.7%) |
| Pathologic M stage (n) |  |  |
| M0 | 320 (97.3%) | 323 (97.0%) |
| M1 | 9 (2.7%) | 10 (3.0%) |
| Borrmann classification (n) |  |  |
| I | 20 (6.0%) | 27 (8.1%) |
| II | 114 (34.7%) | 102 (30.6%) |
| III | 145 (44.1%) | 163 (49.0%) |
| IV | 33 (10.0%) | 26 (7.8%) |
| NA | 17 (5.2%) | 15 (4.5%) |
| Gastrectomy (n) |  |  |
| Distal | 158 (48.0%) | 159 (47.7%) |
| Proximal | 14 (4.3%) | 14 (4.2%) |
| Total | 157 (47.7%) | 160 (48.1%) |
| Adjuvant chemotherapy (n) |  |  |
| SOX | 246 (74.8%) | 274 (82.3%) |
| S-1 | 30 (9.1%) | 32 (9.6%) |
| Other | 22 (6.7%) | 11 (3.3%) |
| No | 31 (9.4%) | 16 (4.8%) |

*SD* standard deviation, *BMI* body mass index (calculated as weight in kilograms divided by height in meters squared), *EIPL* extensive intraoperative peritoneal lavage, *SOX* S-1 plus oxaliplatin, *NA* not available.

**Table 2 Site of first tumor recurrence.**

|  | Surgery alone (n = 329) | Surgery + EIPL (n = 333) | P-value |
|---|---|---|---|
| Overall[a] | 91 (27.7%) | 79 (23.7%) | 0.25 |
| Peritoneum | 39 (11.9%) | 34 (10.2%) | 0.50 |
| Lymph nodes | 23 (7.0%) | 14 (4.2%) | 0.12 |
| Liver | 19 (5.8%) | 25 (7.5%) | 0.37 |
| Lung | 7 (2.1%) | 10 (3.0%) | 0.48 |
| Local | 9 (2.7%) | 4 (1.2%) | 0.17 |
| Other organs | 7 (2.1%) | 10 (3.0%) | 0.48 |

*EIPL* extensive intraoperative peritoneal lavage.
[a]Patients may be included in more than one category of recurrence. $\chi^2$ or Fisher exact test was used for analyses (two-sided).

pathologic N stage. Among the 132 patients with stage N1, the 3-year OS rates were 67.2% (95% CI 55.4–79.0) in the surgery alone group and 84.2% (95% CI 75.7–92.8) in the surgery plus EIPL group (Fig. 3b). Conversely, among the 158 patients with stage N2, the 3-year OS rates were 79.3% (95% CI 70.5–88.0) in the surgery alone group and 68.4% (95% CI 58.0–78.9) in the surgery plus EIPL group (Supplementary Fig. 3a). The HR for death in the surgery and EIPL group is 0.39 (95% CI 0.21–0.75, $p = 0.003$) for patients with stage N1 and 1.82 (95% CI 1.02–3.24, $p = 0.04$) for patients with stage N2. The subgroup analyses of 3-year DFS rates also showed that patients with stage N1, not stage N2, could benefit from EIPL (HR 0.44, 95% CI 0.24–0.81, $p = 0.006$) (Fig. 3c) (Supplementary Fig. 2 and Fig. 3b).

## Discussion

SEIPLUS is a randomized controlled trial to address the survival benefit of EIPL alone with the primary endpoint of OS. Previously, we found eligible patients assigned to the surgery plus EIPL group exhibited reduced 30-days mortality (0 patients) compared with those assigned to the surgery alone group (five patients, 1.9%) ($P = 0.02$). The 30-days morbidity was 17.0% (46 patients) and 11.1% (31 patients) in the surgery alone and EIPL groups, respectively ($P = 0.04$)[16]. No improvement was found in OS or DFS with surgery plus EIPL as compared with surgery alone.

Recently, Japanese investigators have also assessed the value of the EIPL as precautionary measure for PM in CCOG1102 trial. Comparing to that in the control group, they found that the incidence rate of morbidity and mortality tends to decrease in the surgery plus EIPL group. Moreover, in CCOG1102 study, DFS tends to be better in the EIPL group for patients with postoperative infectious complication[14]. These results are similar to the current SEIPLUS study. Indeed, we have reported the short-term results of SEIPLUS study and found that EIPL was helpful to reduce postoperative short-term complications and wound pain[16].

Different from Kuramoto M's report, our study did not confirm the survival benefit of EIPL in locally AGC after 3 years of follow-up[12]. The possible reasons are as follows. First, the entry conditions of the two studies were different. In Kuramoto M's study, only patients with positive intraperitoneal free cancer cells were included. In our study, most of patients were diagnosed as curable AGC with negative peritoneal cytology. Second, a large number of the patients in the current SEIPLUS study received S1-based adjuvant chemotherapy. Given that ACTS-GC trial demonstrated that postoperative treatment with S-1 could significantly decrease PM from 15.8 to 11.2% for AGC patients, the effective treatment of adjuvant chemotherapy might weaken the role of local treatment of EIPL[20,21].

and peritoneal recurrence-free survival in all analyzed patients were shown in Fig. 2. In the post-hoc sensitivity analysis comparing surgery alone with surgery plus EIPL, we excluded 112 patients with pathologic T1, T2, or M1 disease. The OS (HR 0.95, 95% CI 0.73–1.24; log-rank $p = 0.71$), DFS (HR 0.86, 95% CI 0.67–1.10, log-rank $p = 0.23$), and peritoneal recurrence-free survival (HR 0.94, 95% CI 0.72–1.23, log-rank $p = 0.66$) in 550 patients with T3/4NxM0 disease did not differ significantly between the surgery alone and surgery plus EIPL groups (Supplementary Fig. 1). The results of sensitivity analysis were consistent with the ITT results.

**Ad hoc analyses on patients' subgroups by stage.** In addition, the subgroup analyses of 3-year OS rates based on patient and tumor characteristics was presented in Fig. 3a. We found no significant interaction between treatment effect of EIPL and

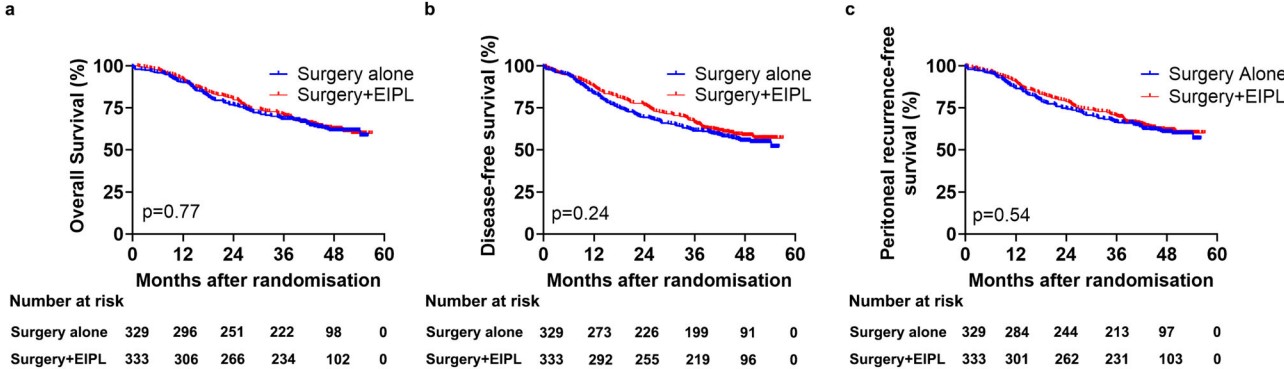

**Fig. 2 Kaplan–Meier estimates of survival curves for the surgery alone and surgery+EIPL groups.** Kaplan–Meier estimates of overall survival (**a**), disease-free survival (**b**), and peritoneal recurrence-free survival (**c**) in all randomized patients by treatment group. Log-rank test was used to compare both curves. EIPL extensive intraoperative peritoneal lavage.

Surprisingly, in the subgroup analyses, 3-year OS rate of patients with N1 disease was better in the surgery plus EIPL group (84.2%) than the surgery alone group (67.2%), whereas in patients with N2 disease, OS rate in the surgery plus EIPL group (68.4%) were worse than the surgery alone group (79.3%). No significant difference was found in patients with N3 disease. The subgroup result needs cautious interpretation and validation in subsequent studies[22,23]. We speculate that the underlying mechanism is related to the metastasis of cancer cells caused by lymph node dissection. Usually, during the process of lymph node dissection, the destruction of lymphatic vessels could lead to tumor implantation[10,24]. Even for 14.3–26.7% of patients with non-serosa-invasive gastric cancer, free cancer cells were found in the lavage fluid after lymphadenectomy[10]. In this case, EIPL could remove these local implant metastases caused by regional lymph node destruction. Conversely, for extensive lymph node metastasis, which often indicates the presence of distant micrometastasis, EIPL fails to prevent the final PM.

Recently, EXPEL group has released their preliminary results, including patients undergoing open and laparoscopic surgery[25]. However, in view of the great controversy of laparoscopic surgery in advanced gastric cancer, we did not include the patients who underwent laparoscopic surgery in our SEIPLUS study. In particular, the reliability of laparoscopic total gastrectomy has not been proved by clinical studies. In fact, laparoscopic treatment of advanced gastric cancer has not been officially recommended as standard operation by NCCN and other guidelines. Moreover, in the narrow abdominal cavity of laparoscopic approach, it's extremely difficult for surgeon to perform regular EIPL, including flushing and full suction.

The major limitation of our study is that 12.1% of patients did not receive S-1-based adjuvant chemotherapy, which might partly impair the quality of the study. Additionally, our study was performed in China, raising questions about the generalizability of the trial findings to other geographic regions.

In summary, although we have found EIPL can decrease postoperative short-term complications, there is no 3-year OS or DFS benefit from EIPL in Chinese patients with AGC.

## Methods

**Study design and participants.** SEIPLUS trial was designed as a phase 3, multicenter, prospective randomized study. Recently, the study design and methods have been published in detail[16]. From April 2016 to November 2017, we enrolled patients with locally AGC from the 11 tertiary hospitals. All participating hospitals are tertiary hospitals in China. The preoperative staging modality included ultrasound gastroscopy and multidetector computed tomography (CT) scans. The intraoperative inclusion criteria were the presence of a stage cT3/4NxM0 tumor according to the macroscopic appearance of exploratory laparotomy. Patients were excluded if they had any of the following criteria: previous neoadjuvant chemotherapy or radiotherapy; positive peritoneal cytology; peritoneal dissemination, distant lymph nodes, ovary, liver, lung, brain, and bone metastases; massive ascites or cachexia; participating in any other clinical trials currently; severe cardiovascular, respiratory, kidney, liver, and mental disease and diabetes; and poor compliance for adjuvant chemotherapy. All candidates provided written informed consent. The ethics committees of every participating center provided ethical approval for the trial. The study protocol accompanying this manuscript (Supplementary Note 1) is a translation of the original study protocol that was amended on October 2020 to include the secondary outcome 3-year peritoneal recurrence-free survival. This amended study protocol was approved by the institutional review board of all the participant hospitals. The full list of participating hospitals was available in Supplementary Note 1. The trial was registered at ClinicalTrials.gov (NCT02745509).

**Randomization and masking.** Patients were assigned (1:1) to the surgery alone or EIPL plus surgery group with a block size of four. Randomization was stratified by each participating hospital. Allocation was executed by sealed opaque envelopes that contained the information of the random number and the procedure to which patients were assigned. The envelopes were opened after meticulous exploratory laparotomy. Patients were masked to treatment assignment. Body mass index (BMI) is calculated by taking a person's weight, in kilograms, divided by their height, in meters squared[26]. The American Joint Committee on Cancer (AJCC) Gastric Cancer Staging version 7 was used for Tumor-Node-Metastasis (TNM) staging. The T3/4 were defined as follows: T3, tumor penetrates the serosa without invasion of the adjacent structures; T4, tumor invades serosa visceral peritoneum or adjacent structures. The N stage were defined as follows: N0, no nodal involvement; N1, metastases in 1 to 2 regional lymph nodes; N2, metastases in 3–6 regional lymph nodes; N3, metastases in more than 7 regional lymph nodes[27]. The Borrmann classification according to gross type of tumor, confirmed by postoperative macroscopic pathologic examination, was defined as follows: type I, polypoid tumors, sharply demarcated from the surrounding mucosa; type II, ulcerated carcinomas with sharply demarcated and raised margins; type III, ulcerated carcinomas without definite limits, infiltrating into the surrounding wall; type IV, diffusely infiltrating carcinomas in which ulceration usually is not a marked feature[28]. All analyses were performed on ITT principle, except for a sensitivity analysis.

**Procedures.** All patients received open curative D2 gastrectomy with margin-negative resection. A total, proximal or distal gastrectomy was done depending on the primary tumor location. The techniques of anastomosis were selected by the preference of surgeons. The laparoscopic gastrectomy was not allowed in the present trial. D2 lymphadenectomy was defined according to the Japanese gastric cancer treatment guidelines[29].

Next, the physiological saline was heated up to 37 °C in incubator. In the surgery alone group, patients received conventional peritoneal lavage using no more than 3 L of warm physiological saline (1 liters of saline solution for 2–3 times). In the surgery plus EIPL group, patients received EIPL using 10 L or more of warm physiological saline (1 L for at least 10 times) after the curative gastrectomy. Every time, the contents of peritoneal cavity was stirred and washed sufficiently, and in turn the fluid was aspirated entirely.

After operation, all participants were recommended to receive eight 3-week cycles of intravenous oxaliplatin (100 mg/m² on day 1 of each cycle) and oral S-1 (40 mg/m² twice daily on days 1–14 followed by 1 week of rest of each cycle). Serious or life-threatening adverse events were managed by dose reductions or interruptions.

**a**

| | 3-year OS rate (%) (95%CI) | | | | Log-rank | Interaction |
|---|---|---|---|---|---|---|
| | Surgery alone | Surgery plus EIPL | | HR (95%CI) | p value | p value |
| **Age, years** | | | | | | 0.86 |
| <60 | 76.4 (69.5-83.3) | 76.9 (70.0-83.7) | | 1.00 (0.65-1.53) | 0.99 | |
| >60 | 62.3 (55.2-69.3) | 65.6 (58.8-72.4) | | 0.95 (0.70-1.30) | 0.76 | |
| **Sex** | | | | | | 0.83 |
| Male | 67.1 (61.1-73.0) | 69.5 (63.6-75.4) | | 1.00 (0.74-1.33) | 0.98 | |
| Female | 72.6 (63.1-82.2) | 73.0 (64.3-81.7) | | 0.95 (0.58-1.56) | 0.84 | |
| **Smoking status** | | | | | | 0.45 |
| No | 69.2 (63.2-75.2) | 69.2 (63.2-75.2) | | 1.04 (0.77-1.41) | 0.79 | |
| Yes | 66.9 (57.8-76.0) | 73.4 (65.1-81.7) | | 0.85 (0.54-1.33) | 0.47 | |
| **BMI** | | | | | | 0.43 |
| <22 | 68.9 (61.8-75.9) | 67.7 (60.4-75.0) | | 1.08 (0.76-1.53) | 0.68 | |
| >22 | 68.1 (60.9-75.3) | 73.1 (66.6-79.7) | | 0.88 (0.62-1.26) | 0.48 | |
| **Tumor location** | | | | | | 0.23 |
| Upper 1/3 | 63.6 (54.2-73.1) | 66.0 (56.6-75.4) | | 1.05 (0.68-1.61) | 0.83 | |
| Middle 1/3 | 71.6 (62.2-81.0) | 66.3 (56.3-76.3) | | 1.43 (0.87-2.36) | 0.16 | |
| Lower 1/3 | 70.5 (62.7-78.4) | 77.1 (70.2-84.1) | | 0.71 (0.47-1.07) | 0.10 | |
| Total | 63.6 (35.2-92.1) | 60.0 (29.6-90.4) | | 0.99 (0.25-3.98) | 0.99 | |
| **Tumor size, cm** | | | | | | 0.76 |
| <5 | 72.9 (67.0-78.8) | 76.2 (70.4-81.9) | | 0.99 (0.72-1.38) | 0.97 | |
| >5 | 59.6 (50.4-68.8) | 60.5 (51.7-69.3) | | 0.92 (0.62-1.35) | 0.66 | |
| **Pathologic T stage** | | | | | | 0.58 |
| T1/2 | 91.5 (83.5-99.5) | 97.7 (95.8-99.7) | | 0.83 (0.25-2.71) | 0.75 | |
| T3 | 77.0 (67.4-86.6) | 76.3 (66.8-85.9) | | 0.84 (0.47-1.50) | 0.55 | |
| T4 | 60.2 (53.5-66.9) | 62.9 (56.4-69.4) | | 1.01 (0.76-1.34) | 0.94 | |
| **Pathologic N stage** | | | | | | 0.65 |
| N0 | 87.5 (79.9-95.1) | 88.0 (80.6-95.4) | | 1.28 (0.61-2.71) | 0.52 | |
| N1 | 67.2 (55.4-79.0) | 84.2 (75.7-92.8) | | 0.39 (0.21-0.75) | 0.003 | |
| N2 | 79.3 (70.5-88.0) | 68.4 (58.0-78.9) | | 1.82 (1.02-3.24) | 0.04 | |
| N3 | 49.0 (39.8-58.3) | 51.8 (42.5-61.0) | | 0.95 (0.67-1.34) | 0.77 | |
| **Pathologic M stage** | | | | | | 0.79 |
| M0 | 69.5 (64.4-74.6) | 71.8 (66.9-76.7) | | 0.97 (0.75-1.26) | 0.83 | |
| M1 | 33.3 (2.5-64.1) | 30.0 (1.6-58.4) | | 0.79 (0.27-2.38) | 0.68 | |
| **Borrmann classification** | | | | | | 0.20 |
| I | 65.0 (44.1-85.9) | 74.1 (57.5-90.6) | | 0.66 (0.25-1.76) | 0.41 | |
| II | 74.3 (66.2-82.4) | 79.4 (71.6-87.3) | | 0.73 (0.44-1.22) | 0.23 | |
| III | 61.4 (53.5-69.3) | 64.4 (57.0-71.8) | | 1.04 (0.75-1.45) | 0.81 | |
| IV | 66.7 (50.6-82.8) | 57.7 (38.7-76.7) | | 1.15 (0.53-2.50) | 0.72 | |

0.25  0.5  1  2  3 4

◄— Surgery plus EIPL better      Surgery alone better —►

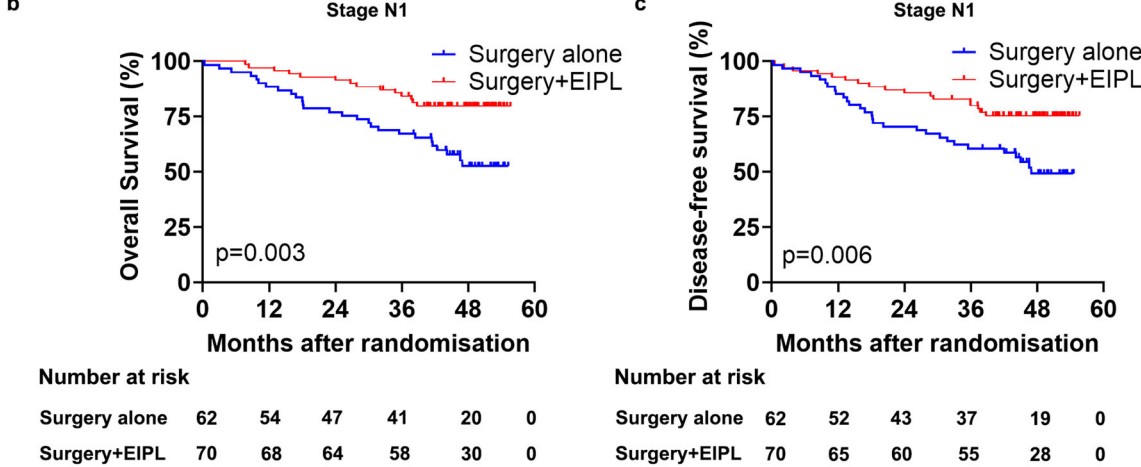

**b** Stage N1

— Surgery alone
— Surgery+EIPL

p=0.003

Overall Survival (%)
Months after randomisation

Number at risk

| | | | | | | |
|---|---|---|---|---|---|---|
| Surgery alone | 62 | 54 | 47 | 41 | 20 | 0 |
| Surgery+EIPL | 70 | 68 | 64 | 58 | 30 | 0 |

**c** Stage N1

— Surgery alone
— Surgery+EIPL

p=0.006

Disease-free survival (%)
Months after randomisation

Number at risk

| | | | | | | |
|---|---|---|---|---|---|---|
| Surgery alone | 62 | 52 | 43 | 37 | 19 | 0 |
| Surgery+EIPL | 70 | 65 | 60 | 55 | 28 | 0 |

**Fig. 3 Subgroup analyses. a** Subgroup analyses of overall survival according to the clinicopathological characteristics of the patients ($n = 662$). Data are plotted as HR value (red squares) with the corresponding two-sided 95% CI (error bars) based on the Cox regression method. **b**, **c** Kaplan–Meier estimates of overall survival (**b**) and disease-free survival (**c**) in stage N1 patients by treatment group. Log-rank test was used to compare both curves. EIPL extensive intraoperative peritoneal lavage, OS overall survival, HR hazard ratio, 95% CI 95% confidence interval, BMI body mass index.

**Outcomes**. The primary endpoint was 3-year OS. The secondary endpoints included 3-year disease-free survival (DFS), 3-year peritoneal recurrence-free survival (reported in this manuscript), and 30-day postoperative complication and mortality. In the present manuscript, the 30-day postoperative complication and mortality was not showed for previously reported[16]. 3-year peritoneal recurrence-free survival is a crucial outcome to assess the PM. Thus, based on the already published study protocol, we added the 3-year peritoneal recurrence-free survival as one of the secondary outcomes in follow-up[16]. OS is calculated from the date of randomization to the date of death whatever the cause or date of censoring. DFS was calculated from the day of randomization to the day of disease recurrence or death whatever the cause or date of censoring. Peritoneal recurrence-free survival was calculated from the day of randomization to the day of peritoneal recurrence or death irrespective of cause or date of censoring. The cutoff date for the final analysis was November 1, 2020.

All patients were followed up regularly and registered. During the first 3 years after surgery, all patients were followed up by medical history, physical examination, and blood testing with tumor markers every 3 months for the first 2 years and subsequently every 6 months. Chest and abdominal CT was performed every 6 months. Upper gastrointestinal endoscopy was performed annually. Recurrence including peritoneum, lymph nodes, liver, lung, et al. was diagnosed by medical history and physical examination combined with imaging, cytology, or histology (if clinically needed). The nongastric cancer-related death was mainly diagnosed by comorbidity-related examination and confirmed by specialist physician.

**Statistical analysis**. This trial was designed to assess the superiority of surgery plus EIPL compared with surgery alone for the OS. We planned to detect an increase in 3-year OS benefit from 60% for the surgery alone group to 71% for the surgery plus EIPL group with 80% power at a two-sided 5% significant level, requiring 254 patients in each group. We planned to recruit patients for 2 years and follow-up patients for 3 years.

Continuous variables were presented as mean and standard deviation. Categorical data were presented as counts and percentages and compared using the $\chi^2$ or Fisher exact test. In the primary analyses, study treatment groups were compared with the stratified log-rank test. The analyses were stratified on center. The Kaplan–Meier method was used to estimate OS, DFS, and peritoneal recurrence-free survival. The survival endpoints were described by their rate at specific time points with 95% confidence intervals (95% CI). Regression analysis was estimated using the Cox proportional hazards model after confirmation of the proportional hazard assumption and described by Hazard ratios (HR) and 95% CI. The proportional hazards assumption was tested by time-dependent covariate analysis.

The statistical analysis was performed with SPSS software, version 17.0 (IBM Corporation, Armonk, NY, USA) and R for Windows, version 3.3.3 (http://www.r-project.org/). We used the library "forestplot" to draw forest plots in R. A P-value less than 0.05 was considered statistically significant.

**Ad hoc analyses**. The ad hoc analyses consisted on test of interaction and sub-group survival analyses, for the purpose of searching potential AGC patients who could benefit from EIPL. Interaction was tested by the likelihood ratio for according to baseline characteristics including age, sex, smoking status, BMI, tumor location, tumor size, T stage, N stage, and Borrmann classification. The subgroup survival analyses included the Cox regression, log-rank test, and Kaplan–Meier method. The Cox regression and log-rank test were used to identify the prognostic factors for OS and DFS. Kaplan–Meier method was used to estimate OS and DFS on patients' subgroups by stage.

**Reporting summary**. Further information on research design is available in the Nature Research Reporting Summary linked to this article.

## Data availability
The data of patient characteristics, therapies, and survival information have been deposited in Research Data Deposit (RDD) public platform (http://www.researchdata.org.cn), with the Approval Number as RDDA2021002105. The data could be obtained via contacting the RDD platform (rdd@sysucc.org.cn). The clinical protocol is available for review in the Supplementary Note 1. Source data are provided with this paper.

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

## Acknowledgements

This was an investigator-initiated trial partly funded by the 308 Project of Sun Yat-sen University Cancer Center and Science and Technology Planning Project of Guangdong Province of China (Grant Serial Number:2016A020215089). The sponsor had no role in design and conduct of the study; collection, management, analysis, and interpretation of the data; and preparation, review, or approval of the manuscript or the decision to submit for publication.

## Author contributions

D.X. designed the study. J.G., A.X., X.S., X.Z., Y.X., H.R., Y.Z., R.Z., L.C., T.Z., G.L. and H.X. collected, analyzed, and interpreted data, and helped to write the report.

## Competing interests

The authors declare no competing interests.
