## [Peer Review File · Nature Communications]

Three-year outcomes of the randomized phase III SEIPLUS Trial of Extensive Intraoperative Peritoneal Lavage for Locally Advanced Gastric CancerREVIEWER COMMENTS

Reviewer #1 (Remarks to the Author):

The authors report on the long term results of a trial of EIPL as an adjuvant therapy delivered at the time of resection (done via open technique) of gastric cancer for locally advanced (T3/4NxM0). The report finds no advantage for the EIPL except for the ~20% of patients with N1 disease. This is an update of the previously published short term results which found decreased pain after surgery with EIPL.

The manuscript is well written and of interest to readers.

I have several questions for the authors:

1. What was the yield of the EGD performed annually for follow up?
2. On what was based an 11% overall survivorship benefit at 3 years of follow up based for the power calculation for the study?
3. The DFS difference of 5.1%, with an absolute OS of 2% and a 2.8% benefit for EIPL begs the question of an underpowered study. How confident are the authors that DFS would not be a better primary endpoint?
4. Would the authors suggest EIPL (a low cost, quick adjuvant) in any setting?

Reviewer #2 (Remarks to the Author):

The authors should be congratulated to carry out an RCT in surgery that enrolled 660 patients in 18 months. No difference neither in OS nor in DFS is reported. Peritoneal recurrence-free survival is described as an outcome but is not reported.

My main concern is on the analysis of the data that do not stick to the consort with a high rate of post-randomization exclusion, that strongly limits the expected benefit of the randomization. The risk of unmeasured confounding factors is then high. The presentation of the flow chart is not clear as the timing of randomization is not adequately presented. One has to read the previous paper (Jama surgery) to have the correct figure.

Point by point comments:

- Abstract:

- o The sample size in the experimental arm is not correct
- o The subgroup analysis is an overstatement (see corresponding comments) and should not be reported in the abstract)
- o This is not an intent to treat analysis as more than 1/6th of the randomized patients have not been analysed.

- Introduction:

- o Line 96: the cited references do not provide mean survival but median survival

- Methods:

- o "the 11 tertiary hospitals of China": does it mean that all tertiary hospitals in China participated?
- o Line 148: The reader would need more information regarding the inclusion criteria described as "poor compliance". As the surgery is the first treatment, this is not clear to what treatment compliance applies.
- o Line 158: "sealed envelopes" is not a commonly used mean to randomize patients nowadays as there is a risk that the investigators do not respect the sequence of envelopes. Do the sponsor monitored and controlled the envelopes in each center?
- o ITT implies that ALL randomized patients are analysed in the group they were assigned irrespective of the protocol deviation (Hollis and Campbell (1999). "What is meant by intention to treat analysis? Survey of published randomised controlled trials")
- o Outcomes:

- ♣ How was defined overall survival? All death or cancer-related death?

- ♣ It is recommended that the OS is calculated from the date of randomization to the date of death whatever the cause or date of censoring.
- ♣ Please provide the cut-off date
- ♣ Peritoneal recurrence-free survival : how were counted recurrences outside of the peritoneum ?
- o Statistical analysis:
 - ♣ Was the analyses stratified or adjusted on the center?
 - ♣ Cox proportional hazard is introduced but in the results, a mixed-effect (frailty?) model is reported. Please correct and if frailty model, provide information regarding the frailty distribution.
 - ♣ How was the proportional hazard assumption checked?
 - ♣ A stratified log-rank test would have been more appropriate as it avoids strong assumptions to compare both groups.
 - ♣ Package for forest plots is introduced, but no sub-group analysis is indicated. In the protocol, there is no description of the subgroups.
- Results:
 - o Make clear that 112 patients were excluded after randomization. Correct the flow chart accordingly. Provide exclusion per treatment arm.
 - o L265: How were non disease related death counted? How was it assessed?
 - o Provide median survival in addition to the 3 year OS or 3-year DFS.
 - o No data on time to peritoneal relapse is provided.
 - o A table on the 30-days mortality and morbidity is missing. If the figures are extracted from the previous publication (Jama surgery), then cite the results in the discussion
 - o Was the log-rank test used for the subgroup only as it does not seem to have been used for the primary analysis. Justify why 2 different approaches?
 - o Subgroups were not introduced. How those subgroups were defined? They are not listed (not even mentioned) in the protocol. Due to the multiple testing, under the hypothesis of no effect, there is a 40% risk of false positive results.
 - o The results in the N1 group that goes in the opposite direction than the treatment effect in the N2 group, that appears also different than the effect in the N3 group is very likely a false positive one explained by mere random fluctuations. This subgroup is clearly overstated.
 - o Indicate how interaction was tested (this is not described in the method section)
- Discussion:
 - o To much emphasize on the N1 subgroup. The conclusions are then not supported by the data.
 - o Absence of ITT should be stressed out as a major limitation.

Reviewer #1 (Remarks to the Author):

The authors report on the long-term results of a trial of EIPL as an adjuvant therapy delivered at the time of resection (done via open technique) of gastric cancer for locally advanced (T3/4NxM0). The report finds no advantage for the EIPL except for the ~20% of patients with N1 disease. This is an update of the previously published short term results which found decreased pain after surgery with EIPL.

The manuscript is well written and of interest to readers.

--We thank the reviewer for recognizing the importance and novelty of this research study and for providing very constructive comments. We have addressed each comment and we believe that our study has been tremendously strengthened.

I have several questions for the authors:

Q1. What was the yield of the EGD performed annually for follow up?

--Thank so much for your useful comments. EGD was mainly used to diagnose the local recurrence and make biopsy. In the present study, 13 patients suffered from local recurrence, who were confirmed by EGD.

Q2. On what was based an 11% overall survivorship benefit at 3 years of follow up based for the power calculation for the study?

--Based on expected improvements in 3-year disease-free survival of 15%

chosen by Misawa^[1] and 3-year overall survival of 10% chosen by Kim^[2], a survival benefit of 11% was postulated for the EIPL group in our study.

- [1] Misawa K, Mochizuki Y, Ohashi N, et al. A randomized phase III trial exploring the prognostic value of extensive intraoperative peritoneal lavage in addition to standard treatment for resectable advanced gastric cancer: CCOG 1102 study. Japanese journal of clinical oncology. 2014;44(1):101-3.
- [2] Kim G, Chen E, Tay AY, et al. Extensive peritoneal lavage after curative gastrectomy for gastric cancer (EXPEL): study protocol of an international multicentre randomised controlled trial. Japanese journal of clinical oncology. 2017;47(2):179-84.

Q3. The DFS difference of 5.1%, with an absolute OS of 2% and a 2.8% benefit for EIPL begs the question of an underpowered study. How confident are the authors that DFS would not be a better primary endpoint?

--We are appreciative of your helpful suggestion. Given the fact that recurrence of gastric cancer is difficult to detect by clinical signs and imaging techniques, especially for the peritoneal metastasis, DFS would not be a better primary endpoint. Indeed, in conventional clinical trials, overall survival as a primary endpoint is considered the gold standard due to its stronger statistical power and easier assessment.

Q4. Would the authors suggest EIPL (a low cost, quick adjuvant) in any setting?

--Currently, EIPL did not improve the 3-year overall survival or disease-free survival in patients with advanced gastric cancer. However, we also found that EIPL can decrease postoperative short-term complications.

Indeed, after the preliminary results were published in JAMA Surg (2019,154(7):610-616), we saw that many surgeons around us began to pay attention to peritoneal lavage. They reflected to us that the surgical complications were significantly reduced, and there are some long-term survival advanced cases with EIPL, which was consistent with our findings in this study. Based on these results, we recommend the use of EIPL in the operation of advanced gastric cancer.

Reviewer #2 (Remarks to the Author):

Q1. The authors should be congratulated to carry out an RCT in surgery that enrolled 660 patients in 18 months. No difference neither in OS nor in DFS is reported. Peritoneal recurrence-free survival is described as an outcome but is not reported.

--We appreciate the reviewer for recognizing the importance of our work and providing the comments for us to strengthen our study. Currently, we report the “Peritoneal recurrence-free survival” in the 3rd paragraph of the part of “Results”. The estimated 3-year peritoneal recurrence-free survival rates were 64.2% (95%CI 58.5-69.9) in the surgery alone group and 67.0% (95%CI 61.5-72.5) in the surgery plus EIPL group. The peritoneal recurrence-free survival (HR 0.94, 95%CI 0.72-1.23, log-rank $p=0.66$) in all 550 patients did not differ significantly between the surgery alone and surgery plus EIPL groups.

Q2. My main concern is on the analysis of the data that do not stick to the consort with a high rate of post-randomization exclusion, that strongly limits the expected benefit of the randomization. The risk of unmeasured confounding factors is then high. The presentation of the flow chart is not clear as the timing of randomization is not adequately presented. One has to read the previous paper (Jama surgery) to have the correct figure.

--We are sorry that we did not make it clear. Indeed, a high rate of post-

randomization exclusion limits the expected benefit of the randomization. However, based on “E9 Statistical Principles for Clinical Trials (ICHE9)”, there are a limited number of circumstances that might lead to excluding randomized subjects from the full analysis set, also known as modified intention-to-treat analysis. According to ICHE9, subjects who fail to satisfy an entry criterion may be excluded from the analysis without the possibility of introducing bias under the following circumstances:(1) the entry criterion was measured prior to randomization. (2) the detection of the relevant eligibility violations can be made completely objectively. (3) all subjects receive equal scrutiny for eligibility violations. (4) all detected violations of the particular entry criterion are excluded. Our study meets the points 2-4.

--As gastric cancer patients with stage T1/2 or M1 are hardly benefit from EIPL, accurate staging is important for the present study. However, before the randomization, we cannot get the accurate pathological staging data. Whereas the pathologists, as third part, made the pathological diagnosis objectively and were blinded with the treatment assignment. In fact, in the present study, the number and clinicopathologic characteristics of excluded patients were similar in the two groups.

--Currently, 112 (16.9%) patients were excluded owing to T1, T2 or M1 disease, which could elevate the risk of unmeasured confounding factors

in some ways. This is the limitation of our study, and we have explained it in the paper.

--As advised, the flow chart has been revised.

Point by point comments:

Q3. Abstract:

The sample size in the experimental arm is not correct

--We apologize for our mistakes. There were 279 patients in the surgery plus EIPL group. And we have made a revision in the part of “Abstract”.

Q4. The subgroup analysis is an overstatement (see corresponding comments) and should not be reported in the abstract.

--We agree with the reviewer and the part of subgroup analysis has been deleted in the abstract.

Q5. This is not an intent to treat analysis as more than 1/6th of the randomized patients have not been analyzed.

--The intention-to-treat principle requires all patients in a randomized trial to be analyzed according to their original randomized allocation. In the present study, the modified intent-to-treat analysis instead of intention-to-treat analysis was used. Generally, modified intention-to-

treat analysis, which based on exclusion of patients postrandomization for different reasons, is commonly used in cancer-related clinical trials, such as CeTeG/NOA-09 ^[1] and AVERT trial ^[2].

[1] Herrlinger U, Tzaridis T, Mack F, et al. Lomustine-temozolomide combination therapy versus standard temozolomide therapy in patients with newly diagnosed glioblastoma with methylated MGMT promoter (CeTeG/NOA-09): a randomised, open-label, phase 3 trial. *Lancet*. 2019;393(10172):678-88.

[2] Carrier M, Abou-Nassar K, Mallick R, et al. Apixaban to Prevent Venous Thromboembolism in Patients with Cancer. *N Engl J Med*. 2019;380(8):711-9.

--As suggested, the parts of “Abstract” and “Randomization and masking” have been revised.

Q6. Introduction:

Line 96: the cited references do not provide mean survival but median survival

--Thank you for your valuable comments. As advised, we have made a revision in the part of “introduction”.

Q7. Methods:

“the 11 tertiary hospitals of China”: does it mean that all tertiary hospitals in China participated?

--We are sorry that we don’t describe it clearly. In the present study, “the 11 tertiary hospitals of China” does not mean that all tertiary hospitals in China participated. It means that all participating hospitals are tertiary hospitals in China. And we have made a revision in the part of

“Study design and participants”.

Q8. Line 148: The reader would need more information regarding the inclusion criteria described as “poor compliance”. As the surgery is the first treatment, this is not clear to what treatment compliance applies.

--We apologize that we don't make it clear. Except for surgery, adjuvant chemotherapy has some effect on the patients' survival. Some Chinese patients reject chemotherapy for fear of side effect. Therefore, the compliance mainly applies on adjuvant chemotherapy. And we have added it in the part of “Study design and participants”.

Q9. Line 158: “sealed envelopes” is not a commonly used mean to randomize patients nowadays as there is a risk that the investigators do not respect the sequence of envelopes. Do the sponsor monitored and controlled the envelopes in each center?

--Indeed, sealed envelope randomization has a risk that the investigators do not respect the sequence of envelopes. Therefore, we emphasize and supervise the use of envelopes in each center. All envelopes are also checked by a special person to ensure that they are completely sealed.

--In fact, as an easy and simple approach to make randomization, sealed envelope has been used in many important multicenter clinical studies, such as the GAND-emesis study^[1]. Based on meta-epidemiological study

of 389 randomized trials from 19 systematic reviews and 65 meta-analyses with altering methods of treatment allocation. Herbison P, et al revealed that sealed envelopes with opaque can give adequate concealment [2]. There is no evidence that the use of sealed envelopes with enhancement was different from central randomization.

[1] Ruhlmann CH, Christensen TB, Dohn LH, et al. Efficacy and safety of fosaprepitant for the prevention of nausea and emesis during 5 weeks of chemoradiotherapy for cervical cancer (the GAND-emesis study): a multinational, randomised, placebo-controlled, double-blind, phase 3 trial. *Lancet Oncol.* 2016;17(4):509-518.

[2] Herbison P, Hay-Smith J, Gillespie WJ, Different methods of allocation to groups in randomized trials are associated with different levels of bias. A meta-epidemiological study, *J Clin Epidemiol.* 2011;64(10):1070-5.

Q10. ITT implies that ALL randomized patients are analyzed in the group they were assigned irrespective of the protocol deviation (Hollis and Campbell (1999). "What is meant by intention to treat analysis? Survey of published randomized controlled trials")

--The intention-to-treat principle requires all patients in a randomized trial to be analyzed according to their original randomized allocation. Frequently, a less strict intention-to-treat approach is used commonly referred to as modified intention-to-treat analysis, which is commonly used in cancer-related clinical trials, such as CeTeG/NOA-09 [1] and AVERT trial [2]. As suggested, the parts of “Abstract” and “Randomization and masking” have been revised.

[1] Herrlinger U, Tzaridis T, Mack F, et al. Lomustine-temozolomide combination therapy versus standard temozolomide therapy in patients with newly diagnosed glioblastoma with methylated MGMT promoter (CeTeG/NOA-09): a randomised, open-label, phase 3 trial. *Lancet (London,*

England). 2019;393(10172):678-88.

[2] Carrier M, Abou-Nassar K, Mallick R, et al. Apixaban to Prevent Venous Thromboembolism in Patients with Cancer. N Engl J Med. 2019;380(8):711-9.

Q11. Outcomes:

How was defined overall survival? All death or cancer-related death?

--Overall survival is calculated from the date of randomization to the date of death whatever the cause or date of censoring. It is "all death". As advised, the part of "Outcomes" has been given detailed description.

Q12. It is recommended that the OS is calculated from the date of randomization to the date of death whatever the cause or date of censoring.

--Thank so much for your useful advice. As suggested, the part of "Outcomes" has been revised.

Q13. Please provide the cut-off date.

--Thank for your valuable suggestion. The cut-off date was November 1, 2020. We have added the date in our manuscript.

Q14. Peritoneal recurrence-free survival: how were counted recurrences outside of the peritoneum?

--Peritoneum recurrence was diagnosed by physical examination (including sign of ascites, digital rectal examination, et al) combined with imaging, cytology, or histology. We described it in the 2nd paragraph of

“Outcomes”.

Q15. Statistical analysis:

Were the analyses stratified or adjusted on the center?

--Yes, the analyses were stratified on center. The reason is that the participating hospitals are all tertiary hospitals in China, the difference of diagnosis and treatment level is small, and the homogenization level is high.

--As advised, we added the description in the part of “Statistical analysis”.

Q16. Cox proportional hazard is introduced but in the results, a mixed-effect (frailty?) model is reported. Please correct and if frailty model, provide information regarding the frailty distribution.

--Thank for your helpful advice. The Cox regression model rather than mixed-effect or frailty model was used for analyzing the risk factors of prognosis in the present study. We have corrected this mistake. As advised, the part of “Result” has been revised.

Q17. How was the proportional hazard assumption checked?

--We examined the proportional hazards assumption using time-dependent covariate analysis. The statistical results were as follows. We found none were significant based on a p value threshold of 0.05. As

suggested, the manuscript has been revised.

Risk factor	P value	
	Overall survival	Disease-free survival
Procedure (Surgery alone/Surgery+EIPL)	0.33	0.31
Age (≤60/>60)	0.18	0.82
Sex (Male/Female)	0.21	0.84
Smoking status (No/Yes)	0.34	0.89
BMI (≤22/>22)	0.77	0.06
Tumor location (Upper 1/3/Middle 1/3/Lower 1/3/Total)	0.80	0.83
Tumor size (≤5 vs >5)	0.89	0.67
Pathologic T stage (T3/T4)	0.21	0.99
Pathologic N stage (N0/N1/N2/N3)	0.58	0.67
Borrmann classification (I/II/III/IV)	0.30	0.99

Q18. A stratified log-rank test would have been more appropriate as it avoids strong assumptions to compare both groups.

--We agree with the reviewer and the stratified log-rank test was used in the revised manuscript.

Q19. Package for forest plots is introduced, but no sub-group analysis is indicated. In the protocol, there is no description of the subgroups.

--Thanks for your comments. We revised our manuscript and added the sub-group analysis in the part of “Statistical analysis”.

--As suggested, we have made description of the subgroups, including BMI, Stage T3/T4, Stage N and Borrmann classification in the part of “Randomization and masking”.

Q20. Results:

Make clear that 112 patients were excluded after randomization. Correct the flow chart accordingly. Provide exclusion per treatment arm.

--We apologize to the reviewer for our mistake. We have made a revision in the part of Figure 1.

Q21. L265: How were non disease related death counted? How was it assessed?

--We are sorry we didn't make it clear. The non gastric cancer related deaths mainly included comorbidity of heart, brain and lung and accident. The comorbidity is diagnosed by disease-related examination and confirmed by specialist physician. The information of accidents was obtained from relatives of patients in the follow-up.

--As advised, we have added it in the part of "Outcomes".

Q22. Provide median survival in addition to the 3 year OS or 3-year DFS.

--We appreciate your helpful suggestion. Indeed, median survival is preferably used to present result of survival analysis. Currently, we found that median overall survival was not reached in either treatment group at the cut-off time. As many patients were still alive in the cut-off date, these cases were considered censored. Whereas the proportion of

censored patients was greater than 50%, the median survival time could not be obtained. Therefore, we had to present the 3-year survival rates as a substitute.

--In fact, the similar EXPEL trial ^[1] and CCOG1102 trial ^[2] also only reported the 3-year OS or 3-year DFS.

EXPEL trial

surgery group had died. 3-year overall survival was 77.0% (95% CI 71.4–81.6) for the EIPL group and 76.7% (71.0–81.5) for the standard surgery group. There was no evidence of difference in hazard of death between the two treatments (HR 1.09 [95% CI 0.78–1.52]; $p=0.62$; figure 2A). Median overall survival had not been reached in either treatment group at the time of final interim analysis. The results remained unchanged after

CCOG 1102 trial

DFS and OS are shown in Fig. 3. The 3-year DFS rate was 63.9 (95 per cent c.i. 55.5 to 71.2) per cent in the EIPL group and 59.7 (51.3 to 67.1) per cent in the control group (HR 0.81, 95 per cent c.i. 0.57 to 1.16; $P=0.249$). The 3-year OS rate was 75.0 (67.1 to 81.3) per cent in the EIPL group and 73.7 (65.9 to 80.1) per cent in the control group (HR 0.91, 0.60 to 1.37; $P=0.634$).

- [1] Yang HK, Ji J, Han SU, et al. Extensive peritoneal lavage with saline after curative gastrectomy for gastric cancer (EXPEL): a multicentre randomised controlled trial. *Lancet Gastroenterol Hepatol.* 2021;6(2):120-7.
- [2] Misawa K, Mochizuki Y, Sakai M, et al. Randomized clinical trial of extensive

intraoperative peritoneal lavage versus standard treatment for resectable advanced gastric cancer (CCOG 1102 trial). The British journal of surgery. 2019;106(12):1602-10.

Q23. No data on time to peritoneal relapse is provided.

--We are appreciative of your helpful suggestion. We reported the “Peritoneal recurrence-free survival” in the 3rd paragraph of the part of “Results”. The estimated 3-year peritoneal recurrence-free survival rates were 64.2% (95%CI 58.5-69.9) in the surgery alone group and 67.0% (95%CI 61.5-72.5) in the surgery plus EIPL group. The peritoneal recurrence-free survival (HR 0.94, 95%CI 0.72-1.23, log-rank p=0.66) in all 550 patients did not differ significantly between the surgery alone and surgery plus EIPL groups.

Q24. A table on the 30-days mortality and morbidity is missing. If the figures are extracted from the previous publication (Jama surgery), then cite the results in the discussion.

--Thank so much for your useful comments. In our previous publication, we found patients assigned to the surgery plus EIPL group exhibited reduced 30-days mortality (0 patients) compared with those assigned to the surgery alone group (5 patients, 1.9%) (P=0.02). The 30-days morbidity was 17.0% (46 patients) and 11.1% (31 patients) in the surgery alone and EIPL groups, respectively (P=0.04). As advised, we have made a revision in the part of

“discussion”.

Q25. Was the log-rank test used for the subgroup only as it does not seem to have been used for the primary analysis. Justify why 2 different approaches?

--We apologize for not explaining statistics of the log-rank test clearly in the manuscript. In both primary and subgroup analyses, we used the log-rank test. It was same approach. We have made a revision in the part of “Statistical analysis”.

Q26. Subgroups were not introduced. How those subgroups were defined? They are not listed (not even mentioned) in the protocol. Due to the multiple testing, under the hypothesis of no effect, there is a 40% risk of false positive results.

--Thank for your helpful comments. We have added description of the subgroups, including Body mass index (BMI), Stage T3/T4, Stage N and Borrmann classification in the protocol.

--We totally agree with the reviewer for these suggestions. The subgroup result needs cautious interpretation and validation in subsequent studies. We have deleted the emphasis of the N1 subgroup.

Q27. The results in the N1 group that goes in the opposite direction than

the treatment effect in the N2 group, that appears also different than the effect in the N3 group is very likely a false positive one explained by mere random fluctuations. This subgroup is clearly overstated.

--We appreciate your valuable advice. Interpretation of the results of the subgroup analyses have to be judged carefully. The potential survival benefit of EIPL in patients with N1 stage need to be confirmed by future studies. As suggested, we have made a revision in the manuscript.

Q28. Indicate how interaction was tested (this is not described in the method section).

--We apologized for the missing description of how interaction was tested. In the subgroup analysis, the likelihood ratio was used to test the interaction. We have added the description in the part of “Statistical analysis”.

Q29. Discussion:

To much emphasize on the N1 subgroup. The conclusions are then not supported by the data.

--We totally agree with the reviewer for these suggestions. We have deleted the emphasis of the N1 subgroup.

Q30. Absence of ITT should be stressed out as a major limitation.

--As suggested, we added the absence of ITT as a major limitation.

REVIEWER COMMENTS

Reviewer #2 (Remarks to the Author):

The authors have clarified numerous aspects.

My concern is still the absence of ITT: considering the fraction of excluded patient, the risk of bias (that can be in either way) is high and might modify the overall conclusions.

All recommendations favor the ITT and this analysis could be made. This analysis on the subset of patients (the actual analysis) could be included as a sensitivity analysis to validate the robustness of the results.

Reviewer #2 (Remarks to the Author):

The authors have clarified numerous aspects.

--We thank the reviewer for your precious valuable suggestions. We believe that our study has been tremendously strengthened.

Q1. My concern is still the absence of ITT: considering the fraction of excluded patient, the risk of bias (that can be in either way) is high and might modify the overall conclusions. All recommendations favor the ITT and this analysis could be made. This analysis on the subset of patients (the actual analysis) could be included as a sensitivity analysis to validate the robustness of the results.

-- We appreciate the reviewer for the suggestion. As advised, we followed up the patients who were excluded and added ITT analysis through the latest survival data. The previous analysis (the modified ITT) were included as a sensitivity analysis. We found the results of ITT were similar with the modified ITT analysis, which validated the robustness of our study. As shown in the new manuscript, estimated 3-year OS rates were 68.5% (95%CI 63.5-73.5) in the surgery alone group and 70.6% (95%CI 65.7-75.5) in the surgery plus EIPL group (Fig 1a). Estimated 3-year DFS rates were 61.2% (95% CI 55.9-66.4) in the surgery alone group and 66.0% (95% CI 61.0-71.1) in the surgery plus EIPL group

(Fig 1b). Estimated 3-year peritoneal recurrence-free survival rates were 66.6% (95%CI 61.5-71.8) in the surgery alone group and 70.0% (95%CI 65.0-74.9) in the surgery plus EIPL group (Fig 1c). The OS (HR 0.97, 95% CI 0.76-1.25, log-rank $p=0.83$), DFS (HR 0.88, 95%CI 0.69-1.11, log-rank $p=0.27$) and peritoneal recurrence-free survival (HR 0.93, 95%CI 0.73-1.20, log-rank $p=0.59$) in all 662 patients did not differ significantly between the surgery alone and surgery plus EIPL groups. Accordingly, the manuscript has revised thoroughly.

Fig. 1 Kaplan–Meier estimates of overall survival (a), disease-free survival (b) and peritoneal recurrence-free survival (c) in all randomized patients by treatment group. EIPL, extensive intraoperative peritoneal lavage.

REVIEWER COMMENTS

Reviewer #2 (Remarks to the Author):

I thank the author for doing the ITT analysis; The results are now definitely convincing.